# LUQ: Layerwise Ultra-Low Bit Quantization for Multimodal Large Language Models

**Shubhang Bhatnagar**[*][†]                                             *shubhangb97@gmail.com*
*University of Illinois Urbana-Champaign*

**Andy Xu**[*][†]                                                        *andyxu05@ucla.edu*
*University of California, Los Angeles*

**Kar-Han Tan**                                                          *karhan.tan@ieee.org*
*HP Inc.*

**Narendra Ahuja**                                                       *n-ahuja@illinois.edu*
*University of Illinois Urbana-Champaign*

Reviewed on OpenReview: *https://openreview.net/forum?id=3eK6U6ZiSp*

## Abstract

Large Language Models (LLMs) with multimodal capabilities have revolutionized vision-language tasks, but their deployment often requires huge memory and computational resources. Post-training quantization (PTQ) has successfully compressed language models to as low as 1-bit precision, its effectiveness for multimodal LLMs (MLLMs) remains unexplored. In this paper, we present the first method for ultra-low-bit (<4-bit) quantization of MLLMs. Our analysis reveals that multimodal tokens and intermediate layer activations produced by them exhibit significantly higher entropy compared to text tokens, indicating greater functional complexity that makes MLLMs less tolerant to ultra-low bit quantization. However, this entropy varies significantly across layers, with some layers producing lower-entropy activation distributions that we empirically show can better tolerate ultra-low bit quantization. Existing PTQ methods optimize weight quantization within each layer but apply the same target precision uniformly, ignoring this variation in complexity across layers. Building on this insight, we propose LUQ: Layerwise Ultra-Low Bit Quantization, which characterizes each transformer layer's functional complexity via its output activation entropy and selectively applies ultra-low bit quantization to layers encoding simpler, more compressible functions. We also show that multimodal calibration (image and text tokens) boosts VQA performance in the ultra-low bit regime. Evaluated on LLaVA-1.5 and Qwen-2.5-VL across 9 VQA benchmarks, LUQ models use 40% and 31% less memory than their 4-bit counterparts while exhibiting less than 10% degradation on MME.

## 1 Introduction

Multimodal Large Language Models (MLLMs) (Liu et al., 2024a; Abdin et al., 2024; Team et al., 2024; Achiam et al., 2023; Bai et al., 2025) have achieved remarkable performance on a variety of vision-language tasks, including visual question answering, image captioning, and spatial reasoning. However, these models are extremely resource-intensive, with large open-source models containing billions of parameters, requiring substantial memory making their deployment expensive (Zhu et al., 2024). Model compression techniques, particularly quantization and pruning, have emerged as promising approaches to reduce the computational

---

[*]Equal contribution.
[†]Work done as an intern at HP Inc.

and memory requirements of these models. Quantization (Courbariaux et al., 2016; Frantar et al., 2022; Lin et al., 2024; Yuan et al., 2024) has proven especially effective at reducing model size while maintaining performance. Recent advances (Malinovskii et al., 2024; Huang et al., 2024a) have pushed the boundaries further, achieving compression to even 1-2 bit-widths.

Despite these advances, most research on quantization has focused on language-only LLMs, particularly in the context of post-training quantization (PTQ) methods such as those of Nagel et al. (2020); Chee et al. (2023), which involve calibrating the model on a small dataset without requiring full retraining. However, the impact of PTQ on multimodal performance remains unexplored. Notably, Huang et al. (2024c) report a significant decline in multimodal task performance when multimodal LLMs are quantized to fewer than 4 bits, in stark contrast to the relatively minor performance drops observed in language-only tasks.

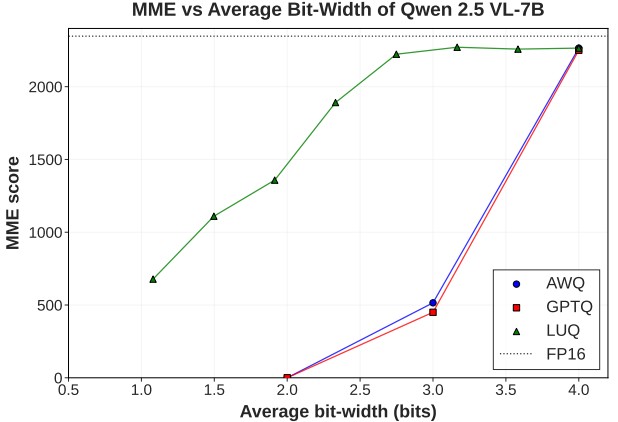
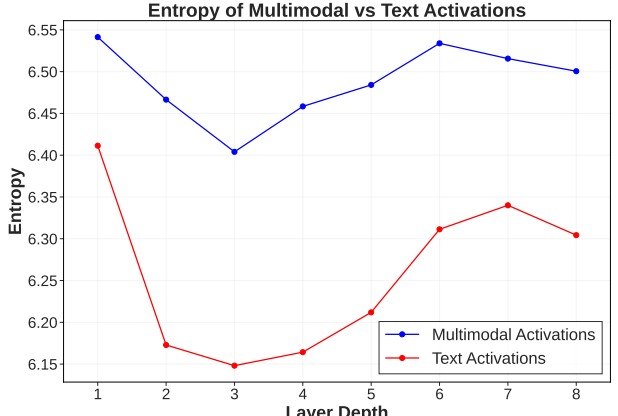

Figure 1: **Performance vs. Compression Trade-off for Qwen 2.5 VL.** Our method, Layerwise Ultra-Low Bit Quantization (LUQ), achieves a better trade-off on the MME benchmark compared to AWQ and GPTQ baselines when used to quantize the multimodal LLM in the ultra-low bit regime.

Figure 2: **Entropy of intermediate activation distributions of Multimodal vs Text only tokens in Qwen 2.5 VL.** Activations produced by multimodal tokens have significantly higher entropy than purely text tokens, potentially explaining poorer resilience of multimodal LLMs to quantization.

To address this gap, we conduct an in-depth study of ultra-low bit (< 4-bit) PTQ for Multimodal LLMs. Our findings confirm that they suffer from a performance collapse on multimodal tasks, frequently generating incoherent outputs in response to paired image-text queries. We hypothesize that this vulnerability stems from the greater functional complexity of multimodal tasks compared to text-only tasks handled by LLMs. To test this hypothesis, we use the entropy of intermediate transformer activations as a proxy for functional complexity and compare the entropy of multimodal tokens against their text-only counterparts. Our analysis on Qwen 2.5 VL (Bai et al., 2025), summarized in Figure 2, reveals that activations of multimodal tokens consistently exhibit higher entropy than those of text tokens, underscoring the increased complexity of processing multimodal inputs. Crucially, however, this complexity is not uniform across the network: different layers exhibit significantly different activation entropy suggesting varying levels of robustness to quantization.

Existing PTQ methods optimize weight quantization within each layer but apply the same target precision uniformly, ignoring this variation in complexity across transformer layers. We propose the first layer-level characterization of quantization robustness for MLLMs, using the entropy of output activations as a proxy for each layer's functional complexity. The intuition is straightforward: a layer whose output collapses to a low-entropy distribution is encoding a simpler, more compressible function, where many weight perturbations map to nearly the same output and can therefore tolerate coarser quantization. Conversely, a layer producing high-entropy outputs relies on finer distinctions among its weights, making it more sensitive to quantization noise. Our experiments show that layers with higher entropy activation distributions are empirically less tolerant to ultra-low bit quantization. We note that our evidence demonstrates entropy is an effective proxy

rather than a strict causal relationship; carefully disentangling it from other correlated properties affecting quantization robustness is an interesting direction for future work.

Based on this characterization, we introduce Layerwise Ultra-Low Bit Quantization (LUQ), a quantization strategy for Multimodal LLMs. LUQ selectively quantizes a subset of network layers to ultra-low bit widths while maintaining 4-bit precision elsewhere. The selection process is driven by a greedy, iterative algorithm that, at each step, quantizes the layer with the lowest activation entropy. This process terminates once a predefined performance threshold on a validation set or a target memory budget is met.

Our approach leverages standard Post-Training Quantization (PTQ) methods for layer-wise quantization, as these methods typically quantize each layer independently. The main benefit of this selective approach is a significantly improved trade-off between model performance and memory footprint compared to standard PTQ methods. This is visualized in Figure 1, which shows that for the Qwen-VL model on the MME benchmark, LUQ consistently establishes a better performance-to-memory frontier than its uniformly quantized counterparts.

To comprehensively validate the effectiveness of our method, we apply LUQ to two widely-used MLLMs: LLaVA-1.5 7B and Qwen-2.5 VL 7B. We conduct evaluations across nine standard Visual Question Answering (VQA) benchmarks, where our experiments show that LUQ achieves a substantial reduction in model size, lowering the average parameter bit-width by 40% for LLaVA-1.5 and 31.5% for Qwen 2.5-VL compared to the 4-bit baseline, while only suffering a small degradation in accuracy.

To summarize our contributions,

- We present the first method for ultra-low bit (<4-bit) quantization of Multimodal LLMs. In contrast to existing PTQ methods that optimize weight importance within layers, we introduce a layer-level functional complexity perspective that characterizes each layer's robustness to quantization.

- We introduce Layerwise Ultra-Low Bit Quantization (LUQ), a novel PTQ approach that instantiates this layer-complexity perspective by selectively quantizing layers based on their output activation entropy, which serves as a proxy for functional complexity and hence quantization robustness. This helps LUQ achieve a better compression-performance trade-off.

- We demonstrate the role of calibration data composition in ultra-low bit quantization, showing that multimodal calibration improves performance over text-only calibration in the ultra-low bit regime.

- We benchmark LUQ on 9 standard VQA benchmarks, showing that LUQ reduces the model size of LLaVA-1.5 and Qwen-2.5 VL by 40% and 31% respectively, compared to their 4-bit counterparts, while incurring a performance degradation of less than 10% on the challenging MME benchmark.

## 2 Related Work

### 2.1 Multimodal LLMs

Recent advances in multimodal large language models (MLLMs) have demonstrated impressive capabilities in understanding and reasoning about visual content alongside text. Most contemporary approaches follow a similar architectural pattern: combining a pre-trained vision encoder (Radford et al., 2021) that processes images into visual tokens that can be consumed by a large language model. This is followed by instruction tuning on multimodal datasets. Liu et al. (2024a) pioneered this approach by connecting a frozen CLIP ViT-L/14 encoder with LLaMA, achieving strong performance through careful instruction tuning. This was followed by similar architectures like Phi (Abdin et al., 2024), Qwen VL (Bai et al., 2025) ,llama 3 (Grattafiori et al., 2024), gemma (Team et al., 2025).

Most of these models have been primarily evaluated on visual question answering (VQA) benchmarks. Standard datasets include VQAv2 (Goyal et al., 2017) and GQA (Hudson & Manning, 2019), TextVQA (Singh et al., 2019) and DocVQA (Mathew et al., 2021)

## 2.2 Quantization of Large Language Models

**Post-Training Quantization (PTQ)**: Recent advances in LLM quantization have explored various approaches including Quantization-Aware Training (QAT) (Liu et al., 2023; Dettmers et al., 2023) and Post-Training Quantization (PTQ) (Frantar et al., 2022; Lin et al., 2024; Shao et al., 2024; Xiao et al., 2023; Lee et al., 2024; Yuan et al., 2023) to reduce model footprints while preserving capabilities. Among these, post-training quantization is particularly efficient, requiring no fine-tuning or access to training data, instead relying on only a small calibration set of data. PTQ methods like GPTQ (Frantar et al., 2022), Omniquant (Shao et al., 2024) and AWQ (Lin et al., 2024) achieve 4-bit compression with minimal accuracy loss compared to their floating point counterparts by optimizing weight distributions and channel-wise scaling factors, with methods like (Lin et al., 2024; Lee et al., 2024) selectively preserving some weights in higher precision. Ultra-low bit approaches such as PB-LLM (Yuan et al., 2024), Slim LLM (Huang et al., 2024b) and BiLLM (Huang et al., 2024a) push compression to 1–3 bits by selectively preserving important weights within a layer, followed by techniques like weight binarization and residual approximation. However, these approaches focus on estimating the importance of parameters compared to other parameters in the layer, unlike LUQ which compares the importance of parameters across layers. A complementary line of work applies randomized Hadamard (orthogonal) transforms before quantization to redistribute channel-wise outliers, enabling more uniform per-channel quantization. Notable examples include QuaRot (Ashkboos et al., 2024) and ResQ (Saxena et al., 2025). These methods are orthogonal to LUQ's layer-selection strategy and could be used as the underlying PTQ method within our framework.

**Layerwise Heterogeneity and Quantization Sensitivity:** Recent works (Wang et al., 2025; Skean et al., 2025) reveal that learned representations vary systematically across layers, with mid layers often concentrating task-relevant information in ways that may affect quantization robustness and tolerance to ultra-low bit precision. Nguyen et al. (2025), though focused on smaller vision-only models, showed that selectively applying different precision levels to different layers yields better accuracy-compression trade-offs. Chang et al. (2025) found that specific "difficult" input tokens can cause large activation outliers in certain layers, leading to significant quantization error. This aligns with our finding that multimodal tokens distributions are less tolerant to quantization. Our work builds on these insights, empirically identifying and exploiting the variance in quantization tolerance across layers specifically for Multimodal LLMs.

**Quantization of MLLMs:** Current works on quantization focus on text-only LLMs, evaluating the effects of quantization on text-only benchmarks. Huang et al. (2024c) extend this evaluation to multimodal benchmarks and show that quantizing multimodal LLMs below 4 bits causes their performance to collapse to nearly 0%, whereas text-only LLMs experience only modest performance drops under the same conditions. PTQ approaches also require a small set of calibration data, helping to obtain scaling factors for the quantized weights. While Ji et al. (2024) examine the impact of different calibration sets on pruning, we are not aware of any similar study for PTQ methods, particularly for multimodal LLMs.

We note that our work targets weight compression of the LLM backbone, which constitutes >95% of parameters in typical MLLMs. Complementary directions such as KV cache quantization (Hooper et al., 2024; Liu et al., 2024c) and vision encoder compression address different memory bottlenecks and could potentially be combined with a LUQ-like strategy for further savings.

# 3 The LUQ Method

## 3.1 Setup and Notation

Let $g_\phi$ and $f_\theta$ denote the vision encoder and language model of the multimodal LLM respectively, parameterized by weights $\phi, \theta$. Any given image-text input pair $(\mathbf{I}, \mathbf{T})$ is converted to a sequence of multimodal tokens denoted by $\mathbf{x}$ using the model tokenizer and $g_\phi$. The input sequence $\mathbf{x}$, with length $N$ multimodal tokens is then passed through $f_\theta$, which is a series of $L$ transformer layers, where each layer $f_{\theta_i}$ $(i \in \{1, ..., L\})$ transforms its input representations:

$$\mathbf{h}_i = f_{\theta_i}(\mathbf{h}_{i-1}) \tag{1}$$

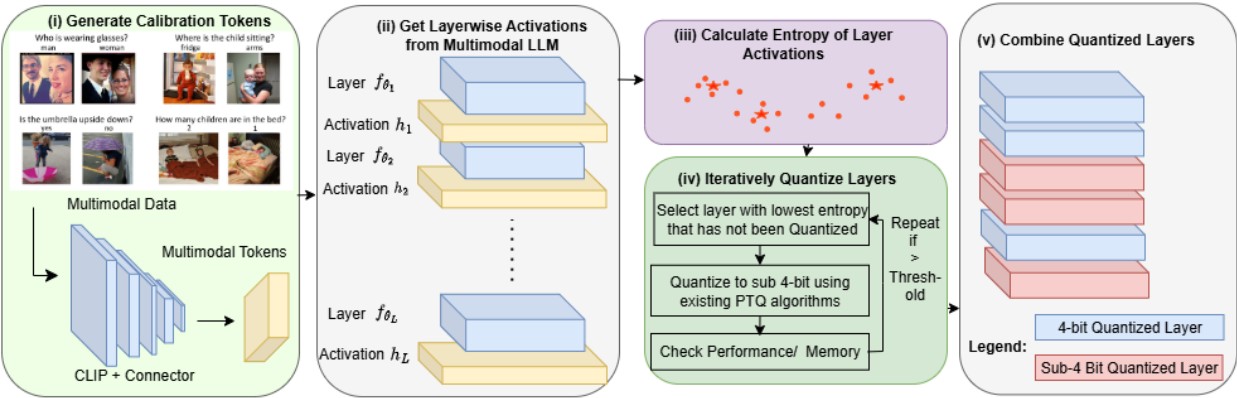

Figure 3: **An overview of our LUQ: Layerwise Ultra-Low Bit Quantization.** (i) Generation of multimodal calibration tokens by passing multimodal data through a CLIP model augmented with a connector to align the modalities; (ii) Extraction of layerwise activations from the multimodal large language model (LLM); (iii) Entropy-based layer selection, where the entropy of activations is calculated to identify the layer most suitable for quantization, prioritizing layers with the lowest entropy to be quantized; (iv) Iterative quantization of layers, where candidate layers are quantized to ultra-low bit precision using existing post-training quantization (PTQ) algorithms. Quantization of each layer is followed by a checking step, where the performance/memory of the candidate LUQ model formed by combining all currently ultra-low bit quantized layers with higher bit layers is compared with a pre-defined memory or performance threshold. The iterations continue if the memory threshold is not met or if the model performs better than the performance threshold and (v) Once the iterative quantization process concludes, the layers quantized to different bit widths are combined back for inference.

where $\mathbf{h}_i$ represents the hidden activations at layer $i$ with $\mathbf{h}_0 = \mathbf{x}$, and $\theta_i$ represents the parameters of the $i$-th layer.

For post-training quantization, we randomly sample a calibration set of $\mathcal{N}$ example sequences denoted by $\mathcal{D}_c$ from the training distribution. Let $X_i$ represent the set of $\mathcal{N}$ intermediate activations produced at the output of layer $f_{\theta_i}$ by $\mathcal{D}_c \forall i$. $X_i$ is used to estimate PTQ quantization parameters and calculate the entropy of intermediate layer output activations for LUQ.

Specifically, given the challenges of quantizing multimodal models, we propose analyzing the complexity of functions encoded by layers $f_{\theta_i}$ in the network, and, by extension, their robustness to quantization. To achieve this, we estimate the entropy of the distribution of $X_i$, their output activations (detailed in Sec. 3.2). The activation entropy $H_i$ for layer $i$ serves as the layer selection metric for our layerwise quantization strategy (detailed in Sec. 3.3), with lower entropy layers being prioritized for quantization. Sec. 3.4 describes the construction of the calibration dataset $\mathcal{D}_c$. Figure 3 gives an overview of our quantization method.

## 3.2 Activation Entropy Estimation

Given the set of layerwise intermediate activations $X_i$, we first convert all elements $x \in X_i$ with dimensions $N \times d$ into $X_i'$, a set of $(\mathcal{N} \times N)$ $d$-dimensional tokens, where $N$ is the sequence length and $d$ is the hidden dimension. We then estimate $H_i$, the entropy of the output distribution, by first estimating the distribution of $X_i'$ using cluster-based discretization. Specifically, we use K-means clustering to partition the $(\mathcal{N} \times N)$ token representations into $K$ distinct clusters. We choose K-means clustering for discretization because transformer activations naturally form clusters in representation space. K-means clustering respects this geometric structure by partitioning the space based on Euclidean proximity to centroids, thereby capturing correlations across all hidden dimensions simultaneously. This stands in contrast to alternatives such as per-dimension histogram binning, which treats each dimension independently, ignoring inter-dimensional correlations and scaling poorly with the hidden dimension d.

A critical parameter in this process is the number of clusters, $K$. This value must be large enough to accurately approximate the activation distribution but not so large that it overfits to sample-specific noise. To determine an optimal $K$ in a principled manner, we perform a rank stability analysis. We compute the entropy-based layer rankings for a range of increasing $K$ values and measure the stability between consecutive rankings using the Normalized Kendall's Tau (Kendall, 1938) distance. We then identify the "elbow" of the resulting stability curve—the point of diminishing returns—using the Kneedle algorithm (Satopaa et al., 2011). This value of $K$ is then used for all subsequent entropy calculations (see Appendix B for the stability curve for Qwen 2.5 VL 7B).

With $K$ determined, we use K-means clustering to partition the $(\mathcal{N} \times N)$ token representations into $K$ distinct clusters:

$$C_i = \text{KMeans}(\mathbf{h}_i, K) \tag{2}$$

where $C_i = \{c_1, ..., c_K\}$ represents the set of cluster centroids.

Given the cluster centroids, let $\phi_i : \mathbf{h}_i \to \{1, ..., K\}$ denote the mapping function that assigns each token activation to its nearest centroid:

$$\phi_i(\mathbf{h}_{i,j}) = argmin_{k \in \{1,...,K\}} \|\mathbf{h}_{i,j} - c_k\|_2 \tag{3}$$

We then compute the empirical probability distribution $P_i$ over the cluster assignments:

$$P_i(k) = \frac{|\{j : \phi_i(\mathbf{h}_{i,j}) = k\}|}{N} \tag{4}$$

Finally, we calculate the entropy $H_i$ of layer $i$'s activations using the standard Shannon entropy formula:

$$H_i = -\sum_{k=1}^{K} P_i(k) \log P_i(k) \tag{5}$$

This entropy provides an estimate of the complexity and diversity in each layer's output representations, which we use as a proxy for robustness to quantization for the layer in our quantization strategy. As shown in Appendix A, the entropy varies significantly across layers, which forms the core motivation for our layer-selective approach.

We note that our vector-quantized entropy metric is inherently robust to orthogonal preconditioners such as the randomized Hadamard transforms used in QuaRot (Ashkboos et al., 2024) and ResQ (Saxena et al., 2025). Because these transforms are orthogonal, they strictly preserve pairwise Euclidean distances between token vectors, leaving K-means cluster assignments and the resulting entropy values unchanged.

### 3.3 Entropy-Guided Progressive Quantization

Given the layer-wise entropy measurements $\{H_1, ..., H_L\}$, we propose an adaptive quantization strategy that prioritizes lower-entropy layers when determining how many can be quantized to ultra-low bit-width while maintaining a target accuracy. The intuition behind this is that layers with lower entropy exhibit more clustered activation patterns containing lower information, suggesting they may be encoding simpler functions that are more amenable to quantization with minimal information loss.

Let $\pi : \{1, ..., L\} \to \{1, ..., L\}$ be a permutation that sorts layers by their entropy in ascending order:

$$H_{\pi(1)} \leq H_{\pi(2)} \leq ... \leq H_{\pi(L)} \tag{6}$$

For a given ultra-low bit PTQ method $Q_{low}$ and a 4-bit PTQ method $Q_{high}$ we progressively quantize layers following this ordering. After quantizing each layer $\pi(i)$, the transformed activations are computed as:

$$\mathbf{h}_i = \begin{cases} f_{Q_{low}(\theta_i)}(\mathbf{h}_{i-1}) & \text{if } i = \pi(j) \text{ for } j \leq k \\ f_{Q_{high}(\theta_i)}(\mathbf{h}_{i-1}) & \text{otherwise} \end{cases} \tag{7}$$

where $k$ represents the current quantization step. Our method is agnostic to the specific quantization approaches, and can leverage any existing PTQ technique, including methods like BiLLM (Huang et al., 2024a) that enable near binary quantization, or more conservative approaches that maintain higher precision.

Instead of conducting validation at each step to determine a $k^*$ such that

$$k^* = \max\{k : \text{Performance} \geq \tau\} \tag{8}$$

where $\tau$ denotes the task specific performance threshold, we can also determine the maximum number of layers that can be quantized to ultra-low bit width using a binary search over $k$ with an upper bound of $L$ and a lower bound of 0, where at each step, we quantize the first $k$ layers in entropy order and evaluate performance on the calibration set. We note that such a binary search is more efficient as it decreases the number of validation runs required, with validation requiring more compute and using a much larger dataset than PTQ for a layer.

The performance metric can be task-specific depending on the benchmark used. This approach allows us to automatically determine the optimal number of layers to quantize while maintaining model quality on the task. Empirically, we find that layers with lower activation entropy can often be quantized to lower bit-widths (e.g., 1-2 bits) without any loss of performance while higher entropy layers may require more precision (e.g., 4 bits) to preserve model performance.

### 3.4 Effect of Calibration Data Selection

Given the distributional difference between text and multimodal token representations in multimodal LLMs observed in Figure 2, we investigate if the choice of calibration data significantly impacts quantization effectiveness. While traditional PTQ methods often use random text samples for calibration, we empirically observe in the results in Sec 4.4.2 that this approach can lead to suboptimal quantization for multimodal LLMs.

We propose to instead use Mixed Modal Calibration, by explicitly sampling both text and visual tokens. Given a calibration budget of $N$ tokens, we construct our calibration set $\mathcal{D}_{\text{cal}}$ by sampling both text tokens $\mathcal{T}$ and multimodal (image) tokens $\mathcal{M}$ from the training distribution:

$$\mathcal{D}_{\text{cal}} = \text{Sample}(\mathcal{T}, (1-\alpha)N) \cup \text{Sample}(\mathcal{M}, \alpha N) \tag{9}$$

where $\alpha$ controls image to text token ratio.

## 4 Experiments and Analysis

In this section, we present empirical validation of our proposed LUQ method along with an analysis of its different components. We begin in Section 4.1 by detailing our experimental setup, including the models, benchmarks, and implementation specifics. In Section 4.2, we present our main results, comparing LUQ quantized LLaVA 1.5 7B (Liu et al., 2024a) and Qwen 2.5 VL 7B (Bai et al., 2025) against state-of-the-art quantization methods across 9 VQA benchmarks and demonstrating its superior performance-memory trade-off for a given threshold. Next, in Section 4.3, we provide a more detailed analysis of the state-of-the-art performance versus memory trade-off achieved by LUQ across a set of different compression rates. Finally, in Section 4.4, we conduct a series of ablation studies to analyze the specific contributions of our key design choices.

| Datasets → Methods ↓ | Avg. Bit width | MME Per. | MME Cog. | MM Bench | Text VQA | VQAv2 | GQA | POPE | Chart QA | Doc QA | Math Vista |
|---|---|---|---|---|---|---|---|---|---|---|---|
| **LLaVA-1.5 7B Backbone** | | | | | | | | | | | |
| FP16 (Baseline) | 16 | 1510 | 350 | 63.4 | 58.2 | 78.5 | 62.0 | 83.2 | - | - | 23.6 |
| GPTQ | 4 | 1450 | 347 | 58.2 | 56.8 | 76.3 | 61.4 | 76.0 | - | - | 20.1 |
| AWQ | 4 | 1456 | 349 | 59.8 | 56.7 | 76.6 | 61.5 | 76.7 | - | - | 20.6 |
| OmniQuant | 4 | 1466 | 318 | 60.1 | 21.0 | 76.6 | 39.2 | 77.6 | - | - | 20.8 |
| SlimLLM | 4 | 1465 | 345 | 63.2 | 25.2 | 76.8 | 37.7 | 80.7 | - | - | 21.2 |
| GPTQ | 3 | 1346 | 273 | 31.2 | 54.1 | 73.5 | 58.8 | 70.5 | - | - | 16.4 |
| OmniQuant | 3 | 1295 | 274 | 46.2 | 16.8 | 72.9 | 15.1 | 17.6 | - | - | 15.2 |
| SlimLLM | 3 | 1429 | 329 | 59.5 | 22.3 | 74.2 | 34.4 | 80.5 | - | - | 18.1 |
| GPTQ* | 2 | 0.0 | 0.0 | 0.0 | 0.0 | 0.0 | 0.0 | 0.0 | - | - | 0.0 |
| AWQ* | 2 | 0.0 | 0.0 | 0.0 | 0.0 | 0.0 | 0.0 | 0.0 | - | - | 0.0 |
| OmniQuant* | 2 | 0.0 | 0.0 | 0.0 | 0.0 | 0.0 | 0.0 | 0.0 | - | - | 0.0 |
| SlimLLM | 2 | 459 | 149 | 0.0 | 0.0 | 0.0 | 0.0 | 9.8 | - | - | 0.0 |
| BiLLM | 1.08 | 561 | 39 | 7.4 | 15.6 | 37.2 | 22.7 | 25.5 | - | - | 3.5 |
| **LUQ 16 layer (Ours)** | **2.54** | **1365 ± 9** | **257 ± 7** | **46.7 ± 0.6** | **53.4 ± 0.4** | **74.9 ± 0.2** | **58.2 ± 0.4** | **74.5 ± 0.7** | **-** | **-** | **18.7 ± 0.9** |
| **Qwen 2.5 VL Backbone** | | | | | | | | | | | |
| FP16 (Baseline) | 16 | 1695 | 640 | 82.6 | 84.9 | 83.5 | 60.5 | 86.1 | 87.3 | 95.7 | 68.2 |
| GPTQ | 4 | 1638 | 610 | 80.2 | 84.2 | 82.6 | 60.1 | 84.8 | 84.1 | 93.4 | 44.8 |
| AWQ | 4 | 1645 | 620 | 80.9 | 84.6 | 82.7 | 60.5 | 85.6 | 84.5 | 93.5 | 46.1 |
| OmniQuant | 4 | 1655 | 610 | 81.3 | 84.5 | 82.9 | 57.5 | 85.8 | 82.9 | 92.6 | 47.5 |
| GPTQ | 3 | 319 | 131 | 34.7 | 79.5 | 81.5 | 53.4 | 82.9 | 61.0 | 89.2 | 21 |
| OmniQuant | 3 | 335 | 135 | 32.3 | 41.3 | 81.6 | 12.8 | 65.1 | 58.5 | 89.7 | 23.5 |
| GPTQ* | 2 | 0.0 | 0.0 | 0.0 | 0.0 | 0.0 | 0.0 | 0.0 | 0.0 | 0.0 | 0.0 |
| AWQ* | 2 | 0.0 | 0.0 | 0.0 | 0.0 | 0.0 | 0.0 | 0.0 | 0.0 | 0.0 | 0.0 |
| OmniQuant* | 2 | 0.0 | 0.0 | 0.0 | 0.0 | 0.0 | 0.0 | 0.0 | 0.0 | 0.0 | 0.0 |
| BiLLM | 1.08 | 638 | 42 | 9.7 | 26.3 | 39.5 | 4.3 | 70.7 | 3.7 | 20.3 | 15.1 |
| **LUQ 12 layer (Ours)** | **2.75** | **1640 ± 10** | **600 ± 5** | **63.7 ± 0.6** | **81.9 ± 0.2** | **79.7 ± 0.4** | **52.9 ± 0.6** | **84.7 ± 0.8** | **68.6 ± 0.5** | **90.5 ± 0.3** | **41.7 ± 0.8** |

Table 1: **Performance of LUQ compared to state-of-the-art PTQ methods on VQA benchmarks for LLaVA-1.5 7B and Qwen 2.5 VL 7B models.** For LLaVA-1.5 7B, LUQ achieves comparable VQA accuracy to 4-bit GPTQ/AWQ while reducing memory requirements by 40%. Similarly, for Qwen 2.5 VL 7B, LUQ maintains strong performance with 31.5% lower memory footprint compared to its 4-bit counterparts. LUQ results are reported as mean ± standard deviation across 3 runs. GPTQ*, AWQ* , and OmniQuant* indicate models with incoherent/gibberish output. Results for LLaVA-1.5 7B on ChartQA and DocQA are excluded as the FP16 baseline performance was too low to enable a meaningful analysis of quantization effects.

## 4.1 Experimental Setup

**Model Architecture:** For our experiments, we use LLaVA-1.5 (Liu et al., 2024a) and Qwen 2.5 VL 7B (Bai et al., 2025), which are widely used in previous works on Multimodal LLMs. The models process images using a CLIP ViT (Radford et al., 2021) visual encoder and project visual features into the language model's embedding space through an MLP projection. We quantize only the language model backbone as it forms the bulk of parameters of the overall network.

**Benchmarks:** We evaluate our quantization method on 9 standard visual question-answering benchmarks: **(1) MME** (Fu et al., 2025), where we report Perception and Cognition scores separately to offer a nuanced view of model capabilities; **(2) MMBench** (Liu et al., 2024b) for comprehensive multi-modal evaluation; **(3) TextVQA** (Singh et al., 2019) for optical character recognition; **(4) VQAv2** (Goyal et al., 2017) for general visual reasoning; **(5) GQA** (Hudson & Manning, 2019) for compositional reasoning; **(6) POPE** (Li et al., 2023) for hallucination evaluation; **(7) ChartQA** (Masry et al., 2022) for question answering on charts and plots; **(8) DocVQA** (Mathew et al., 2021) for visual question answering over document images; and **(9) MathVista** (Lu et al., 2024) for visual mathematical reasoning.

**Calibration Data:** We calibrate the model using a randomly sampled set of 128 sequences, each containing 2048 tokens. The calibration data consists of a 1:1 mixture of text tokens from Wikitext-2 (Merity et al., 2017) and multimodal tokens generated from the TextVQA dataset (Singh et al., 2019), following the findings of our ablation studies (see Section 4.4).

**Choice of PTQ strategy:** We choose to use BiLLM (Huang et al., 2024a) as the PTQ strategy for ultra-low bit quantization due to its state-of-the-art performance in the 1-bit regime. Layers not selected for ultra-low bit quantization are quantized to 4-bits using the standard GPTQ (Frantar et al., 2022) method.

**Implementation Details:** We implement our quantization pipeline in PyTorch. For entropy estimation, we use K-means clustering with $K = 100$ centroids, arrived through the process described in Section 3.2. For all quantization methods (GPTQ and BiLLM), we use a standard block size of 128. To quantize the LLaMA-based layers in LLaVA-1.5 with BiLLM, we adopt the optimized parameters from the original BiLLM paper for LLaMA models (Huang et al., 2024a). All experiments are conducted on a single NVIDIA RTX 5000 GPU with 32GB of VRAM.

### 4.2 Comparison to State-of-the-Art Methods

We compare the VQA performance of the model quantized using LUQ to models quantized using state-of-the-art PTQ methods, including GPTQ, AWQ, OmniQuant(Shao et al., 2024), and SlimLLM (Huang et al., 2024b), to different bit widths to better evaluate the model size to performance trade-off of our method. For this comparison, we create two representative LUQ models to demonstrate its adaptability to different practical deployment scenarios. For LLaVA-1.5, we create a performance-constrained model by setting a threshold of a 100-point decrease on MME Perception relative to the 4-bit GPTQ baseline, which results in quantizing the 16 lowest-entropy layers (2.54 avg. bits per parameter). For Qwen 2.5 VL, we create a memory-constrained model with a hardware motivated budget of 2.5 GB, a hypothetical limit for edge devices (having 4 GB RAM), leading to the quantization of the 12 lowest-entropy layers (2.75 avg. bits per parameter). The results, presented in Table 1 , show that LUQ establishes a new state-of-the-art for ultra-low bit MLLM quantization. For LLaVA-1.5, our LUQ model achieves performance comparable to 4-bit GPTQ and AWQ baselines while being 40% smaller. Specifically, the accuracy drop on TextVQA, VQAv2, and GQA is a modest 3.4%, 1.7%, and 3.2%, respectively, compared to 4-bit GPTQ. On the challenging MME benchmark, the performance cost is only 5.8% on Perception and 25.9% on Cognition, a favorable trade-off for the significant memory savings.

Similarly, for Qwen 2.5 VL, our LUQ model maintains strong performance with a 31.5% smaller memory footprint than its 4-bit counterparts. It remains highly competitive on all benchmarks, showing a drop of only 5.5% and 1.4% on complex benchmarks like DocVQA and POPE respectively. It vastly outperforms the larger 3-bit quantized GPTQ model across all benchmarks despite being 8.3% smaller, highlighting the more favorable balance between quantization and performance achieved by LUQ.

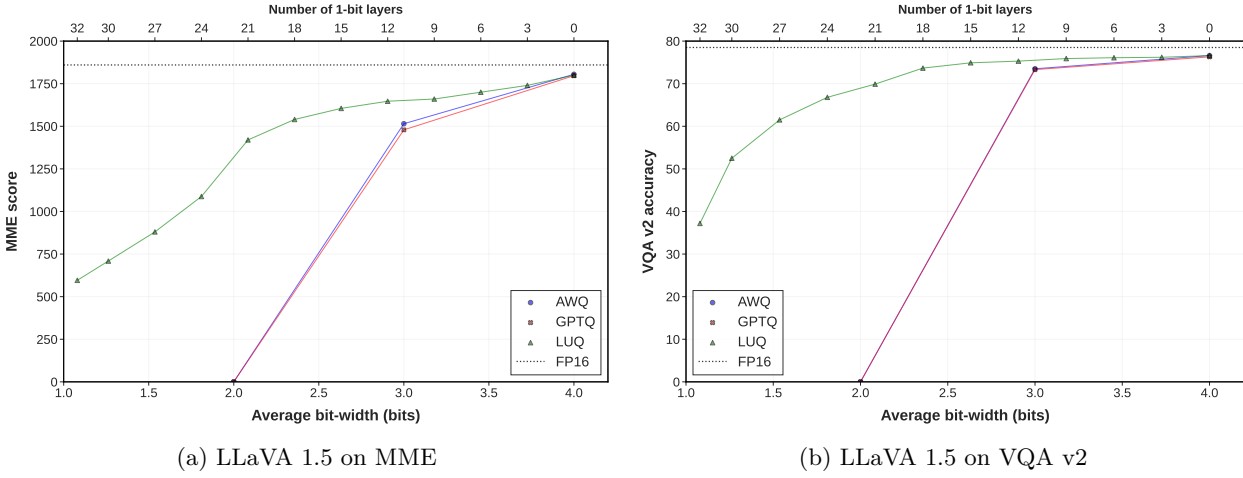

(a) LLaVA 1.5 on MME        (b) LLaVA 1.5 on VQA v2

Figure 4: **Performance versus average bit-width for various post-training quantization methods using LLaVA 1.5 7B. (a)** On the MME benchmark, LUQ significantly outperforms other methods for the LLaVA 1.5 model on the MME benchmark. **(b)** On the VQA v2 benchmark, LUQ maintains high accuracy for LLaVA 1.5 even at aggressive compression rates, whereas baseline methods show a sharp decline in performance.

Standard 2-bit quantization of both Multimodal LLMs with GPTQ, AWQ , and OmniQuant leads to a complete model collapse, producing incoherent outputs. While uniform 1.08-bit BiLLM avoids collapse, its performance is severely degraded, rendering it impractical for real-world applications. LUQ, in contrast, helps navigate this trade-off with more flexibility, delivering compressed but functional models.

### 4.3  Performance vs. Memory Trade-off

To provide a more comprehensive view beyond the fixed points in Table 1, we analyze the continuous trade-off between performance and memory enabled by LUQ. We incrementally quantize more layers to ultra-low bits, starting from the lowest entropy ones, and plot the model's performance against the resulting average bit-width.

Figure 4 illustrates this trade-off for LLaVA-1.5 7B on the MME and VQAv2 benchmarks. The plots show that LUQ facilitates a graceful degradation in performance as the model is compressed. This creates a superior Pareto frontier compared to the discrete, and often distant, performance points offered by standard PTQ methods like GPTQ and AWQ. This flexibility allows selection of the best operating point for specific accuracy and hardware constraints.

We observe a similar trend for Qwen 2.5 VL 7B, as shown in Figure 5 on the DocVQA and ChartQA benchmarks, where LUQ again demonstrates a significantly better performance memory tradeoff as compared to using off the shelf PTQ methods, maintaining high accuracy even at average bit-widths below 3.0 bits where other methods fail catastrophically.

Since any specific performance threshold $\tau$ or memory budget maps directly to a point on these Pareto frontiers, Figures 4 and 5 capture the sensitivity of LUQ to the choice of $\tau$ across benchmarks. Notably, the layer set chosen for LLaVA-1.5 using an MME-Perception threshold (16 layers, 2.54 avg. bits) also degrades gracefully on VQAv2 (Figure 4b), indicating that the Pareto frontiers across benchmarks are broadly consistent.

### 4.4  Ablation Studies

We now ablate LUQ's key design choices. All experiments follow the setup in Section 4.1 unless stated otherwise.

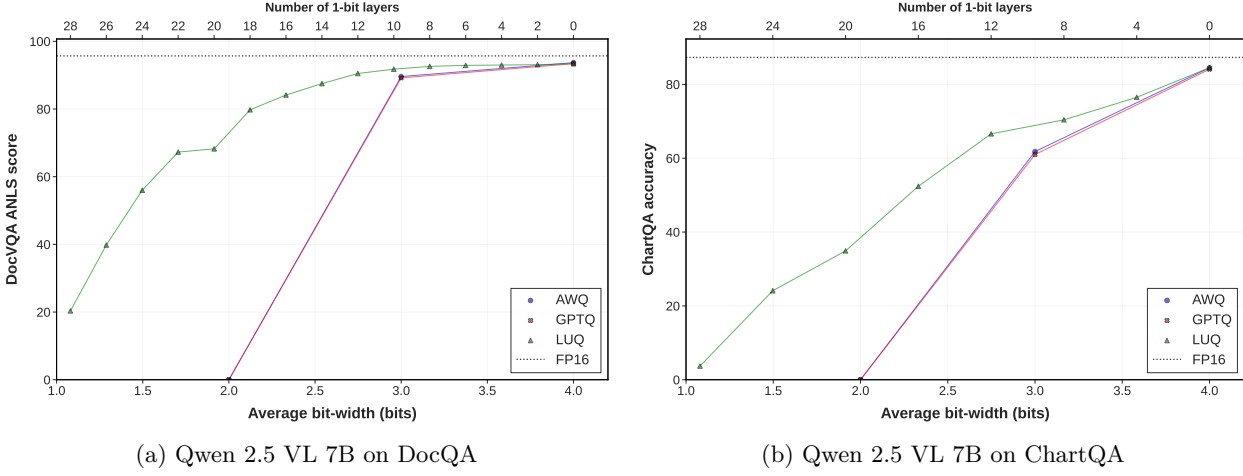

(a) Qwen 2.5 VL 7B on DocQA          (b) Qwen 2.5 VL 7B on ChartQA

Figure 5: **Performance versus average bit-width for Qwen 2.5 VL 7B on (a) DocVQA and (b) ChartQA.** LUQ provides a graceful performance trade-off as more layers are quantized to 1-bit. In contrast, standard PTQ methods like GPTQ and AWQ suffer a catastrophic performance collapse at sub-3-bit compression rates, highlighting LUQ's superior robustness in the ultra-low bit regime.

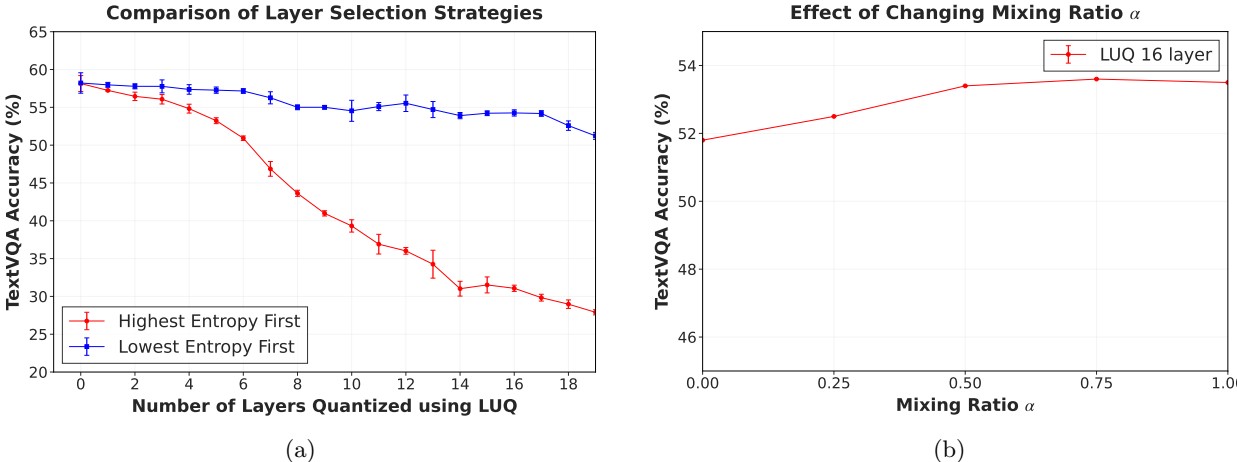

(a)            (b)

Figure 6: **(a) Accuracy on TextVQA for Low Entropy First vs. High Entropy First quantization.** Quantizing low activation entropy layers first preserves performance, while quantizing higher entropy layers first leads to a steep decline in performance. This trend holds for different numbers of layers quantized. **(b) Impact of changing Mixing Ratio $\alpha$ on TextVQA performance.** Even a small $\alpha > 0$ improves performance, but gains saturate as $\alpha$ increases.

### 4.4.1 Entropy as a layer selection metric

**Entropy vs Layer Depth as a selection metric:** We investigate our layer selection strategy, which prioritizes quantizing the lowest entropy layers first, by comparing it against an alternative approach that selects deeper layers (those closer to the model output). This experiment isolates the impact of the selection metric while maintaining all other parameters constant. The choice of layer depth as a selection metric is motivated by recent work (Gromov et al., 2025) focused on layer pruning that empirically showed that deeper transformer layers tend to encode simpler functions compared to shallow layers. To evaluate these approaches, we compare the VQA performance of models with an identical number of quantized layers (16 for LLaVA 1.5 7B and 12 for Qwen 2.5 VL 7B, per Section 4.1) using LUQ, varying only the selection metric. The results, shown in Table 2, demonstrate that while using layer depth as a selection metric yields competitive performance, it is consistently outperformed by LUQ on 5 benchmarks.

**Highest vs Lowest Entropy:** We evaluate the impact of activation entropy-based layer selection by comparing: (1) standard LUQ, which quantizes layers in ascending order of entropy (Low Entropy First), and (2) an inverted strategy that prioritizes layers with highest entropy (High Entropy First) on LLaVA-1.5. Performance is evaluated on TextVQA after each layer is quantized. As shown in Figure 6a, the low entropy first approach significantly outperforms the model quantized using high entropy first across all numbers of quantized layers. The high entropy quantized first model notably undergoes a steep decline in accuracy.

| Methods ↓ | Avg. Bit width | Selection Met. | MME Per. | MME Cog. | TextVQA | VQAv2 | GQA |
|---|---|---|---|---|---|---|---|
| **LLaVA-1.5 7B Backbone** | | | | | | | |
| FP16 (Baseline) | 16 | - | 1510 | 350 | 58.2 | 78.5 | 62.0 |
| LUQ 16 layer | 2.54 | Layer Depth | 1320 | 234 | 53.1 | 74.2 | 57.0 |
| **LUQ 16 layer** | **2.54** | **Entropy** | **1365 ± 9** | **257 ± 7** | **53.4 ± 0.4** | **74.9 ± 0.2** | **58.2 ± 0.4** |
| **Qwen 2.5 VL 7B Backbone** | | | | | | | |
| FP16 (Baseline) | 16 | - | 1695 | 640 | 84.9 | 83.5 | 60.5 |
| LUQ 12 layer | 2.75 | Layer Depth | 1550 | 550 | 79.4 | 74.2 | 51.5 |
| **LUQ 12 layer** | **2.75** | **Entropy** | **1640 ± 10** | **600 ± 5** | **81.9 ± 0.2** | **79.7 ± 0.4** | **52.9 ± 0.6** |

Table 2: **Impact of Selection Metric on Post-Training Quantization (PTQ) for LLaVA-1.5 and Qwen 2.5 VL.** For both models, quantizing layers selected by the lowest entropy consistently outperforms quantizing the deepest layers across all benchmarks, highlighting the effectiveness of the entropy-based selection metric.

| Methods ↓ | Bit Width | Calib. | TextVQA | VQAv2 |
|---|---|---|---|---|
| FP16 (Baseline) | 16 | - | 58.2 | 78.5 |
| GPTQ | 4 | Text | 56.7 | 76.6 |
| GPTQ | 4 | Mix | 56.8 | 76.6 |
| AWQ | 4 | Text | 56.7 | 76.3 |
| AWQ | 4 | Mix | 56.5 | 76.2 |
| GPTQ | 2 | Text | 0 | 0 |
| GPTQ | 2 | Mix | 0 | 0 |
| AWQ | 2 | Text | 0 | 0 |
| AWQ | 2 | Mix | 0 | 0 |
| BiLLM | 1.08 | Text | 11.2 | 35.3 |
| BiLLM | 1.08 | Mix | 15.6 | 37.2 |
| **LUQ 16 layer** | **2.54** | Text | **51.8** | **71.1** |
| **LUQ 16 layer** | **2.54** | Mix | **53.4** | **74.9** |

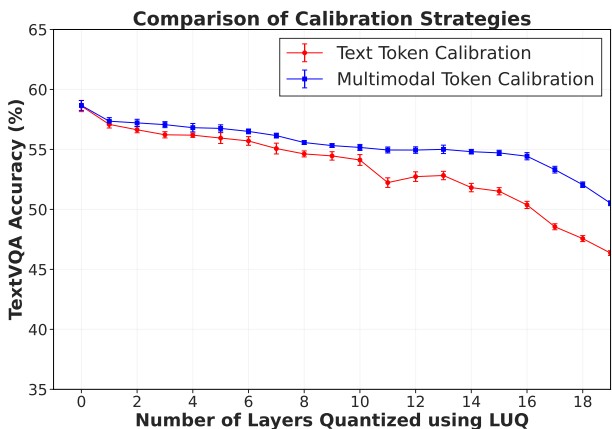

Table 3: **Impact of calibration data composition (multimodal mix vs. text-only tokens) on post-training quantization (PTQ) of LLaVA-1.5.** PTQ methods quantizing the model to less than 4-bits (LUQ and BiLLM) show higher VQA performance improvements with mixed multimodal token calibration, in contrast to 4-bit methods which exhibit marginal improvements.

Figure 7: **Effect of mixed multimodal token vs. text only token calibration on LUQ-quantized models across different quantization levels on TextVQA.** Mixed modal calibration outperforms text-only calibration across all quantized layers, with the performance gap widening as more layers are quantized.

When 16 layers are quantized, the Low Entropy First model achieves a 23.2% higher accuracy on TextVQA, highlighting the importance of prioritizing low entropy layers.

### 4.4.2 Effect of Mixed Modal calibration

**Text Tokens vs Mixed Multimodal Tokens:** We empirically evaluate the effect of calibration data on the multimodal performance of PTQ, by comparing the traditional text-tokens-only approach to our approach of using a mix of multimodal tokens. Our experiments on LLaVA-1.5 reveal that the impact of calibration tokens is dependent on the quantization bit-width. For 4-bit quantization, using a mix of multimodal tokens for calibration achieves comparable performance to using only text tokens for calibration, with only marginal improvements due to the addition of image tokens (Table 3, Block 1). However, for sub-4-bit quantization methods like LUQ and BiLLM, using mixed multimodal tokens for calibration improves VQA performance compared to text-only calibration, as seen in Table 3 Blocks 3 and 4. These results suggest that use of multimodal tokens for calibration becomes increasingly helpful as quantization becomes more aggressive.

**Effect of Mixed Modal Calibration on Varying Quantization Extents:** We evaluate the effect of using multimodal tokens as calibration data on models quantized to different extents using our iterative strategy. Specifically, we varied the number of layers of LLaVA 1.5 7B quantized using LUQ between [0,25] while quantizing models separately using text tokens and multimodal tokens, measuring performance at each point on the TextVQA dataset. Across all configurations, mixed calibration demonstrated better results compared to text-only calibration. For instance, as shown in Figure 7, when quantizing 16 layers using LUQ, mixed calibration achieved 4% higher accuracy on TextVQA compared to text-only calibration. The gap in performance also increased as we increased the proportion of layers quantized using LUQ.

**Effect of Changing Mixing Ratio $\alpha$:** Finally, varying the mixing ratio $\alpha \in \{0, 0.25, 0.5, 0.75, 1.0\}$ for the 16-layer LUQ model (Figure 6b) shows that even a small $\alpha > 0$ yields significant improvement over text-only calibration, with gains saturating at higher $\alpha$.

## 5 Conclusion

In this work, we present the first method for ultra-low bit (<4-bit) post-training quantization of multimodal LLMs. Our analysis reveals that certain layers exhibit lower-entropy activation distributions and can better tolerate ultra-low bit quantization. Based on this, we propose LUQ, a layerwise quantization strategy that selectively applies ultra-low bit precision to more tolerant layers, combined with mixed multimodal calibration

that further improves performance in this regime. Evaluations against multiple PTQ baselines on 9 VQA benchmarks show that LUQ reduces the average bit-width by 40% and 31% over 4-bit counterparts with modest accuracy loss, and that these gains translate to real inference speedups. Extending LUQ to joint weight-and-activation quantization is a promising direction for future work.

## 6 Limitations

LUQ's effectiveness is inherently constrained by the base PTQ method's quality, and it depends on the availability of sufficient validation data and diverse multimodal calibration tokens. Additionally, activation entropy is only an empirical proxy for function complexity; more refined measures could yield better insights into quantization resilience.

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

## A    Entropy of Qwen 2.5 VL Activations

Figure 8 illustrates the core motivation for our proposed Layerwise Ultra-Low Bit Quantization (LUQ) strategy. The entropy values shown were calculated on activations collected using the input data and model settings described in Section 4.1. We observe that the Shannon entropy of intermediate activations is not uniform across the depth of the network. Instead, it exhibits significant variance, with certain layers (particularly in the middle of the network) having much higher entropy than others. This suggests that layers have different levels of representational complexity and, therefore, may have varying resilience to the information loss introduced by aggressive quantization. The layer rankings derived from these entropy values were used for all layer-selection experiments reported in Section 4.

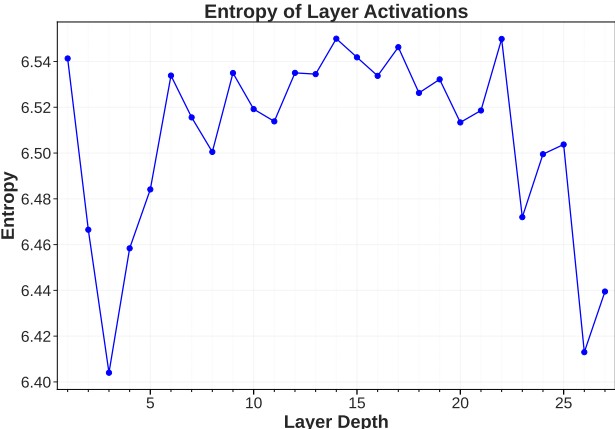

Figure 8: **Variation in Entropy of Intermediate Activation of Qwen 2.5 Vl 7B with Layer Depth.** The Entropy of intermediate activations calculated using the process described Section 3.2 varies significantly with layer depth. This variance motivates our approach of selectively applying ultra-low bit quantization to lower-entropy layers, which we hypothesize are more robust to information loss.

## B    Selection and Validation of Cluster Count ($K$)

The discretization of layer activations via K-means clustering is central to our entropy estimation. A critical hyperparameter in this process is the number of clusters, $K$. We determine the optimal $K$ by analyzing the stability of layer rankings. In B.2 we verify the robustness of downstream task performance to variation in $K$ beyond the selected stability point.

### B.1    Layer Ranking Stability Analysis

To ensure a stable and reliable estimation of layer-wise entropy, we perform a rank stability analysis to select the number of clusters, $K$, for the K-means algorithm. This analysis was conducted on the same set of activations detailed in Section A, generated according to the experimental setup in Section 4.1.

As shown in Figure 9, we plot the Normalized Kendall Tau distance (Kendall, 1938) between the layer rankings generated by consecutively sampled values of $K$. The Normalized Kendall Tau distance quantifies the disagreement between two rankings. The plot shows that as $K$ increases, the layer ordering stabilizes, and the distance metric approaches zero. We use the Kneedle algorithm (Satopaa et al., 2011) to identify the "elbow" of this curve, which represents the point of diminishing returns where increasing $K$ no longer significantly changes the layer ranking. For our experiments, this analysis yielded $K = 100$.

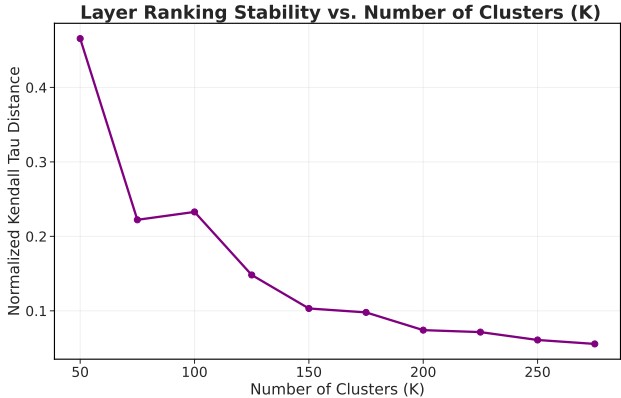

Figure 9: **Stability of the layer ordering $\pi$, defined in Equation 7 (ascending order of layer activation entropy) with variation in number of clusters used for K-Means clustering.** We measure the stability of the layer ordering $\pi_k$ using the Kendall Tau distance.

## B.2 Downstream Performance Sensitivity

To further ensure that our method is not overly sensitive to the specific choice of $K$ beyond the stability point identified above ($K = 100$), we conducted a sensitivity analysis on the downstream task performance. We utilized LUQ to quantize the Qwen 2.5 VL 7B model using the experimental setup described in Section 4.1, varying $K$ from the stability point up to $K = 250$.

Table 4 reports the performance on the MME benchmark. The results demonstrate that once the layer ordering has stabilized, the specific choice of $K$ has a negligible impact on model accuracy. The performance variance across different $K$ values is minimal, particularly when compared to the significant degradation observed in standard 3-bit and 4-bit baselines. This confirms that $K = 100$ is a robust choice for our entropy-guided quantization strategy.

Table 4: Sensitivity of MME performance to cluster count $K$. The consistency of scores for $K \geq 100$ confirms that LUQ is robust to variations in clustering granularity once rank stability is achieved.

| Method | Avg. Bit-Width | MME Perception | MME Cognition |
|---|---|---|---|
| GPTQ (4-bit) | 4.00 | 1645 | 620 |
| GPTQ (3-bit) | 3.00 | 319 | 131 |
| LUQ ($K = 100$) | 2.75 | 1640 | 600 |
| LUQ ($K = 125$) | 2.75 | 1626 | 596 |
| LUQ ($K = 150$) | 2.75 | 1643 | 602 |
| LUQ ($K = 200$) | 2.75 | 1642 | 602 |
| LUQ ($K = 250$) | 2.75 | 1637 | 594 |

## C Benchmark Evaluation Details

**1) MME:** For the MME benchmark (Fu et al., 2025), we use the standard validation set to evaluate both perception and cognition capabilities. The final score is a sum of accuracy scores across sub-tasks, presented as two separate scores for Perception and Cognition.

**2) MMBench:** We evaluate on the dev split of MMBench (English, V1.1)(Liu et al., 2024b). Performance is reported as the overall accuracy.

**3) TextVQA:** We use the validation set of the TextVQA dataset (Singh et al., 2019), which contains 5,000 questions. The evaluation metric is the standard VQA accuracy, which measures the model's ability to answer questions based on textual information within the image.

**4) VQAv2:** For the VQAv2 benchmark (Goyal et al., 2017), we report results on the 'test-dev' split. The evaluation is performed by submitting results to the official evaluation server. We report the standard VQA accuracy metric.

**5) GQA:** We evaluate on the testdev balanced split of the GQA dataset (Hudson & Manning, 2019). This split is a balanced subset of the full test set designed for efficient and robust offline evaluation. We report the overall accuracy.

**6) POPE:** To evaluate object hallucination, we use the validation set from the POPE benchmark (Li et al., 2023), which consists of binary yes/no questions. We report the F1-score as our metric.

**7) ChartQA:** We use the human-annotated test set of ChartQA (Masry et al., 2022). The task involves answering questions about data presented in charts. We report the "relaxed accuracy," which allows for minor formatting differences in the answers.

**8) DocVQA:** For DocVQA (Mathew et al., 2021), we evaluate on the validation set, which comprises 5,349 questions over 1,291 document images. We report the Average Normalized Levenshtein Similarity (ANLS) as the evaluation metric.

**9) MathVista:** We use the 'testmini' split of the MathVista benchmark (Lu et al., 2024) for efficient and robust evaluation. This split is designed to be a representative sample of the full test set. We report the overall accuracy across all question types.

## D   Real-World Deployment and Latency Analysis

While our primary evaluation focuses on the memory compression-performance trade-off as is done by past quantization works, here we address the practical feasibility of deploying LUQ models on commodity hardware.

### D.1   Architectural Compatibility

A key advantage of LUQ is that it only mandates **inter-layer** mixed precision rather than *intra-layer* mixed precision. Unlike methods that require complex kernels to handle different precisions within a single weight matrix, LUQ can use a uniform precision for each layer. This allows the inference engine to utilize optimized, homogeneous kernels for each layer sequentially, avoiding the overhead of frequent context switching or custom matrix multiplication implementations. Consequently, LUQ models can be deployed using existing inference frameworks that support per-layer quantization definitions.

### D.2   Inference Speed Evaluation

| Model Configuration | Intel i7-13620H | AMD Threadripper | Memory Usage |
|---|---|---|---|
| FP16 (Baseline) | $0.2 \pm 0.0$ | $4.9 \pm 0.1$ | 14.5 GB |
| Q4_K_M (Standard 4-bit) | $4.8 \pm 0.3$ | $14.1 \pm 0.2$ | 4.4 GB |
| **LUQ (Mixed Precision)** | $\mathbf{9.0 \pm 0.3}$ | $\mathbf{18.7 \pm 0.1}$ | **3.4 GB** |

Table 5: Inference throughput (tokens/sec) comparison on CPU hardware using `llama.cpp`. Results are averaged over 10 generation runs.

To quantify the practical speed gains, we evaluate the generation latency of Qwen 2.5 VL 7B using `llama.cpp`(Gerganov, 2023), a widely used inference engine optimized for CPU execution. We utilized

the exact layer configuration determined in Section 4.1, mapping our quantization targets to the nearest supported kernels in the engine:

- **Ultra-low bit layers:** Mapped to `IQ1_M` (approx. 1.75 bpw).

- **High precision layers:** Mapped to `Q4_K_M` ( 4-bit).

We measured the generation throughput (tokens/second) on two distinct hardware profiles: a consumer-grade Intel i7-13620H laptop CPU and a workstation-class AMD Ryzen Threadripper PRO 7965WX. Table 5 compares the LUQ configuration against FP16 and standard 4-bit quantization (`Q4_K_M`). LUQ delivers significant speedups over the 4-bit baseline on both platforms while reducing memory usage by approximately 23%.

