# OpenReview forum: "LUQ: Layerwise Ultra-Low Bit Quantization for Multimodal Large Language Models"
_TMLR — Accepted by TMLR_

### Review · Reviewer_egN1 · 2025-10-30

**Summary Of Contributions:**

This paper provides the first in-depth study on ultra-low bit (<4-bit) post-training quantization (PTQ) for Multimodal Large Language Models (MLLMs). The paper first analyzes and reveals the reason for the sharp performance degradation of MLLMs in this setting, which is that the activations from multimodal tokens have significantly higher statistical variance and entropy compared to text-only tokens. Based on this finding, the paper proposes a novel selective quantization strategy—LUQ (Layerwise Ultra-Low Bit Quantization). This method identifies and prioritizes layers for quantization by calculating the entropy of each layer's activations, selecting those more robust to compression (i.e., lower entropy) for ultra-low bit quantization while maintaining 4-bit precision for other layers. Furthermore, the paper demonstrates that using mixed multimodal (image and text) tokens for PTQ calibration in the ultra-low bit regime effectively improves VQA performance. Experimental results show that LUQ achieves significant memory compression on LLaVA-1.5 and Qwen-2.5-VL models, reducing memory footprint by 40% and 31% respectively compared to their 4-bit baselines, while keeping the performance degradation on the MME benchmark within 10%.

$\textbf{Strengths}$

$\textbf{Intuitive Methodological Basis:}$ The paper's core contribution is proposing the use of "activation entropy" as a proxy metric for the quantization robustness of different layers in MLLMs. This idea is intuitive: layers with lower entropy (i.e., more clustered activation patterns) encode simpler functions and are therefore more tolerant of aggressive quantization.

$\textbf{Clear Experimental Validation:}$ The paper's experiments clearly support its claims. In the sub-3-bit regime, LUQ avoids the "performance collapse" seen in baseline methods and demonstrates a superior performance-compression trade-off curve on multiple benchmarks like MME and VQAv2. Key ablation studies also validate the effectiveness of its methodology.

$\textbf{Valuable Additional Finding:}$ The paper also provides an important practical insight: in the ultra-low bit (<4-bit) quantization of MLLMs, using mixed multimodal tokens (image and text) for calibration yields better VQA performance than the traditional method of using only text tokens.

$\textbf{Weaknesses}$

$\textbf{Potential Sensitivity to Hyperparameters:}$ The paper's proposed entropy estimation method relies on K-means clustering, which introduces a key hyperparameter: the number of clusters $K$. Although the authors use the Kneedle algorithm to select $K=100$, the paper does not sufficiently discuss the sensitivity of the model's final performance to changes in the value of $K$. It remains unclear how robust the layer ranking and quantization results are to the choice of $K$.

$\textbf{Lack of Discussion on Practical Deployment Complexity:}$ The LUQ method ultimately produces a mixed-precision model. While this significantly saves memory, the paper does not discuss the potential engineering challenges and the impact on inference latency when deploying this heterogeneous precision on actual hardware. Efficiently executing mixed-precision operations often requires custom compute kernels to handle switches between different bit-widths, which could offset some of the gains.

**Audience:**

Yes

**Audience Explanation:**

Model compression, inference efficiency, and the deployment of large models are active and high-interest research areas. This paper precisely investigates the challenging problem of ultra-low bit (<4-bit) quantization for MLLMs.

Therefore, for researchers and engineers focusing on model quantization techniques and efficient inference, this paper's findings (especially the analysis of multimodal activation entropy) and the proposed LUQ strategy are attractive.

Although the method itself is more of an effective strategy built upon existing PTQ algorithms rather than a fundamental algorithmic innovation, the empirical analysis and the approach it provides for solving the MLLM quantization bottleneck are of reference value to this specific audience.

**Claims And Evidence:**

Yes

**Claims Explanation:**

At a theoretical level, the paper attributes the quantization vulnerability of MLLMs to the high entropy of multimodal activations and provides intuitive data comparisons. Its core hypothesis—that activation entropy is a proxy for quantization robustness—is reasonable, although this assumption is based on intuition rather than rigorous theoretical derivation.

At an experimental level, the paper provides strong and clear evidence for this hypothesis. Key ablation studies demonstrate that prioritizing the quantization of low-entropy layers (the core strategy of LUQ) is indeed far superior to quantizing high-entropy layers. Furthermore, in comparisons with baselines, LUQ clearly shows its superiority in the sub-3-bit regime, avoiding the performance collapse of other methods and providing a smoother performance-compression trade-off curve.

Overall, despite the lack of deeper theoretical analysis, the paper's empirical evidence is accurate and convincing.

**Requested Changes:**

$\textbf{Add sensitivity analysis for the hyperparameter $K$ (Strengthening):}$ The paper's core method relies on K-means clustering to estimate activation entropy, where the number of clusters $K$ is a key hyperparameter. It is recommended to add an experiment in the appendix or ablation studies to analyze the sensitivity of the model's final performance to variations in the value of $K$.

$\textbf{Discuss or evaluate actual inference latency (Strengthening):}$ The paper currently focuses on memory compression. As LUQ produces a mixed-precision model, it is recommended to add a section to the paper discussing the inference latency of this mixed-precision model on actual hardware to provide a more comprehensive efficiency evaluation.

---

> ### Author Response · Authors · 2025-12-08
> **Response to Reviewer egN1**
>
> We thank the reviewer for the valuable feedback, addressing  both strengths and weaknesses. Our response addresses the weaknesses
> ## **W1. Potential Sensitivity to the number of clusters $K$**
> We agree that the choice of \(K\) is important. This is why, in the main paper, we selected \(K\) using the Kneedle algorithm (Sec. 3.2) rather than making an arbitrary empirical choice: as explained in Appendix B of the paper, Kneedle algorithm identifies the saturation point in the stability of the layer ranking after which the Kendall Tau distance between layer orderings for different K becomes negligible.
>
> However, we agree that verifying the downstream performance as $K$ varies can strengthen the paper. Therefore, we use LUQ to quantize Qwen 2.5 VL 7B  following the same experimental setup used in Sec. 4.1 while increasing $K$ above the stability point ($K = 100$). We evaluate the resulting quantized model on the MME benchmark.
>
> The results in the Table below show that the performance remains stable across all tested values of \(K\), confirming that once the layer ordering has stabilized, the exact choice of \(K\) has little effect on final accuracy. For context, we include the performance of GPTQ 4-bit and 3-bit baselines, showing that the performance fluctuations from changing \(K\) are far smaller than the gap between LUQ and standard PTQ.
> **We will include these results in the revised manuscript.**
>
>
> **Table: Sensitivity of MME performance to cluster count \(K\), with GPTQ baselines for reference**
> | Method          | Avg. Bit-Width | MME Perception | MME Cognition |
> |----------------------------|----------------:|----------------:|----------------:|
> | **GPTQ (4-bit)**           | 4.00            | 1645            | 620            |
> | **GPTQ (3-bit)**           | 3.00            | 319          | 131            |
> | **LUQ (K = 100)**          | 2.75            | 1640            | 600            |
> | **LUQ (K = 125)**          | 2.75            | 1626            | 596           |
> | **LUQ (K = 150)**          | 2.75            | 1643             | 602            |
> | **LUQ (K = 200)**          | 2.75            | 1642             |  602           |
> | **LUQ (K = 250)**          | 2.75            | 1637             | 594            |
>
>
>
> ## **W2. Discussion on deployment complexity**
> In our manuscript, we followed the standard practice in prior quantization literature (Sec. 2.2), where evaluations focus on memory reduction and accuracy, as latency is heavily influenced by hardware and kernel level factors which are typically chosen independent of the quantization method.  However, we appreciate the reviewer’s concern regarding engineering challenges for mixed-precision inference.
>
> **Clarifying LUQ’s Mixed-Precision Design:** LUQ itself specifies **mixed precision only across layers**, which substantially reduces the engineering burden.  An LUQ model, when using layer homogenous PTQ techniques, does not require custom intra-layer kernels or frequent precision switches within a single compute kernel, the primary source of the overhead noted by the reviewer. Varying precision across layers is compatible with existing inference frameworks and is considerably easier to deploy than methods that mix precisions *inside* a layer. The additional within-layer heterogeneity seen in our experiments arises from using BiLLM as the ultra-low-bit backend in Sec. 4.1, not from LUQ’s design itself.
>
>
> **Additional Latency Measurements:** We agree that providing latency/generation speed measurements under a consistent inference setting can strengthen the paper. We conducted inference-speed experiments using llama.cpp, a widely used inference backend that supports <4-bit quantization for MLLMs. This allows us to evaluate performance without relying on custom kernels.
>
> Specifically, we evaluate Qwen 2.5 VL 7B using the same LUQ layer selections and bit-width assignments as in Sec. 4.1, varying only the quantization formats to those available in llama.cpp. Since llama.cpp does not currently provide BiLLM or GPTQ/AWQ implementations for MLLMs, we adopt the closest available analogues supported by it:
> - **IQ1_M** for the ultra-low-bit LUQ layers
> - **Q4_K_M** for the remaining 4-bit layers
>
> We report generation throughput (output tokens/sec) on 2 CPU platforms under identical conditions averaged across 10 runs. As shown in the table below, LUQ achieves large speedups across all devices.
>
>
> **Table: Inference speed (tok/s) comparison (Qwen 2.5 VL 7B, llama.cpp)**
> | Model Configuration | i7-13620H | AMD Threadripper* | Memory |
> |:--------------------|----------:|-------------------:|-------:|
> | FP16 (baseline) | 0.2 ±  0.0 | 4.9 ± 0.1 | 14.5 GB |
> | Q4_K_M (4-bit) | 4.8 ± 0.3 | 14.1 ± 0.2 | 4.4 GB |
> | **LUQ (12 IQ1_M layers)**| 9 ± 0.3 | 18.7 ± 0.1 | 3.4 GB |
> *\*Hardware:AMD Ryzen Threadripper PRO 7965WX*
>
> **We will incorporate these latency results and the accompanying discussion into the revised manuscript.**

---

### Review · Reviewer_Vjwg · 2026-02-22

**Summary Of Contributions:**

1. The paper proposes an approach to perform mixed precision weight quantization of multimodal LLMs.
2. The paper proposes a layer selection metric based on layer output activation entropy. Layers with lower entropy are quantized more heavily.
3. The paper analyzes the role of calibration data on multi modal LLM quantization.

**Audience:**

Yes

**Audience Explanation:**

The area of LLM efficiency is relevant to TMLR.

**Broader Impact Concerns:**

No concerns.

**Claims And Evidence:**

Yes

**Claims Explanation:**

1. LUQ does mixed precision quantization where it quantizes less sensitive layers more heavily. As a result, LUQ gets better tradeoff between algorithm performance and average bit width compared with AWQ and GPTQ which do uniform precision quantization.
2. Based on ablations demonstrated, quantizing layers with lower entropy to ultra low precision does much better than quantizing layers with higher entropy.
3. Activation entropy based layer selection metric outperforms depth based selection.
4. Incorporating vision tokens into the calibration data improves the performance of LUQ.

**Requested Changes:**

1. Current comparison in Table 1 only comprises of uniform precision quantization approaches. Please compare with more performant mixed precision quantization approaches.
2. Please add a section on applying LUQ to joint Weight and activation quantization.
3. Please provide motivation behind current approach of calculating entropy. Why was Kmeans and Shannon entropy used to calculate entropy as opposed to other approaches.
4. Also add motivation behind the choice of entropy of output activation as a layer selection metric as opposed to entropy of input activation.
5. Recent LLM quantization approaches apply randomized hadamard transform before performing quantization (e.g. Quarot https://arxiv.org/abs/2404.00456, ResQ  https://arxiv.org/abs/2412.14363). How would that impact the layer sensitivity metric in LUQ.

---

> ### Author Response · Authors · 2026-03-17
> **Response to Reviewer Vjwg**
>
> We thank the reviewer for the focused and constructive feedback, and for raising several interesting directions including the connection to Hadamard-based methods (QuaRot, ResQ). We address each requested change below and have incorporated the suggested references into Sec. 2.2 of the revised manuscript.
>
> ### **RC1. Comparison with mixed-precision quantization approaches**
> We note that Table 1 already includes a mixed-precision baseline: BiLLM. We have now also added SlimLLM, another mixed precision approach to Table 1 for LLaVA-1.5 (SlimLLM does not support Qwen 2.5 VL). It also shows a collapse in performance as it approaches 2 bits.
>
> In case the reviewer is referring to cross-layer mixed-precision methods, to the best of our knowledge, no such method exists for MLLMs, LUQ is the first to address this.
>
>
> ### **RC2. Joint weight and activation quantization**
> We appreciate this suggestion. Activation quantization introduces its own set of challenges that are orthogonal to our layer-selection contribution. We believe this is a sufficiently deep topic that it warrants a dedicated study and have noted it as a promising future direction in Sec. 5 of the revised manuscript.
>
> ### **RC3. Motivation for K-means + Shannon entropy**
>
> We thank the reviewer for this question. Entropy estimation requires discretizing the continuous, high-dimensional activation space. We chose K-means because transformer activations naturally form clusters in representation space, making it a geometrically appropriate discretization unlike per-dimension histogram binning, which ignores correlations across the hidden dimensions and scales poorly with dimensionality.
>
> Shannon entropy is then the natural choice given its direct information-theoretic connection to compressibility, which is precisely what quantization exploits. We have added this discussion to Sec. 3.2 of the revised manuscript.
>
> ### **RC4. Output vs. input activation entropy**
> Our goal is to estimate the complexity of the function encoded by each layer to determine its robustness to quantization. Output activation entropy directly measures this: a layer producing a low-entropy (highly clustered) output is performing a simpler transformation that can be approximated with fewer bits, while a layer producing a high-entropy output relies on fine-grained weight distinctions that are destroyed by aggressive quantization. Input activation entropy, by contrast, is simply the output entropy of the preceding layer, and would therefore misattribute one layer's complexity to the next.
>
> ### **RC5. Impact of randomized Hadamard transforms (QuaRot, ResQ)**
>
> We thank the reviewer for this insightful question, and for the references to QuaRot and ResQ. Applying randomized Hadamard transforms like these methods before quantization will not impact LUQ's layer sensitivity metric.
>
> While these transforms redistribute channel-wise variance to remove channel-specific outlier features, our entropy metric (Sec. 3.2) is computed over the full token vector space via K-means clustering, jointly across all channels. Because Hadamard transforms are orthogonal, they strictly preserve pairwise Euclidean distances between all token vectors. Since the distances are invariant, the K-means cluster assignments remain identical, and consequently the discrete probability distribution over clusters and the resulting entropy value are unchanged.
>
> We have added a brief discussion in Sec. 3.2 of the revised manuscript to clarify that our vector-quantized entropy metric is inherently robust to orthogonal preconditioners like those used in QuaRot and ResQ.

---

### Review · Reviewer_VEdf · 2026-03-02

**Summary Of Contributions:**

The paper studies post-training quantization (PTQ) below 4 bits for multimodal LLMs (MLLMs) and reports that uniform ultra-low-bit PTQ (e.g., 2-bit GPTQ/AWQ) often yields incoherent outputs on VQA benchmarks. The authors hypothesize that multimodal processing is less tolerant to aggressive quantization because multimodal token activations have higher complexity, as shown via activation entropy. The proposed method computes a layerwise entropy statistic $H_i$ by discretizing activations via K-means clustering and applying Shannon entropy. It then ranks layers by increasing entropy and progressively quantizes the lowest-entropy layers to ultra-low precision using a base PTQ method, while leaving the remaining layers at 4-bit GPTQ. The number of quantized layers is selected via a performance threshold $\tau$. The authors also study calibration data composition, proposing mixed modal calibration that mixes text and multimodal tokens, and report that mixed calibration helps more in the sub-4-bit regime. Experiments apply the proposed method to LLaVA-1.5 7B and Qwen 2.5-VL 7B and evaluate across 9 VQA benchmarks.

**Audience:**

Yes

**Audience Explanation:**

The paper provides a relatively simple PTQ-based approach to obtain sub-3-bit average weight precision while preserving non-trivial VQA performance, which (to my understanding) is of practical interest for deployment scenarios where 4-bit models remain too large.

**Broader Impact Concerns:**

The work focuses on model compression and could reduce inference resource requirements. Lower deployment costs can also reduce barriers to the misuse of multimodal LLMs, which leads to positive implications for more accessible ML.

**Claims And Evidence:**

Yes

**Claims Explanation:**

1. Table 1 reports 0 across benchmarks for GPTQ 2-bit and AWQ 2-bit for both backbones, with the caption stating these runs produced "incoherent/gibberish output". This supports the paper’s motivation that naive ultra-low-bit PTQ can fail for MLLMs in this setup.

2. LUQ (the proposed method) achieves average bitwidth 2.54 (LLaVA) and 2.75 (Qwen) with non-zero results on all reported benchmarks, and large improvements over uniform 1.08-bit BiLLM on multiple tasks, e.g., Qwen DocVQA 90.5 vs 20.3

3. Table 2 shows selecting layers by entropy outperforms selecting by layer depth for the fixed quantized-layer counts tested. Fig.6a shows lowest entropy first degrades less than highest entropy first on TextVQA as more layers are quantized.

4. Table 3 shows minimal differences between text-only and mixed calibration at 4-bit, but larger improvements for BiLLM and LUQ.

**Requested Changes:**

1. The paper motivates entropy $H_i$ as a proxy for “functional complexity” and uses it to guide quantization order. The evidence provided shows that entropy-based ordering works better than the tested alternatives, but does not establish that entropy is the underlying causal driver of quantization robustness rather than a correlate of other properties (layer depth, activation outliers, activation scale, attention vs MLP behavior, etc.).

2. Some claims are inconsistent. For example,
- The abstract and conclusion claim “first study” on <4-bit MLLM PTQ, and the contributions repeat "first analysis". However, Sec. 2.2 explicitly cites existing studies as extending quantization evaluation to multimodal benchmarks and showing collapse below 4 bits. In this sense, the "first study" claim might not be supported as written.
- Abstract claims "higher statistical variance and entropy" but only entropy is shown. I did not find corresponding variance measurements/plots in the paper. The variance part of the abstract should be removed or backed by results.

3. Table 1 compares only GPTQ, AWQ, and BiLLM, while related work cites additional PTQ methods (e.g., OmniQuant). If those methods cannot be applied in this MLLM setting (or cannot go below 4-bit), the paper should state so explicitly and/or evaluate the strongest applicable alternatives.

4. Table 1 reports "mean performance across 3 runs", but Tables/Figures provide no standard deviations or confidence intervals. Since calibration sampling and K-means can introduce randomness, mean-only reporting is insufficient to judge whether smaller deltas (e.g., Table 2) are meaningful.

5. The method selects the number of quantized layers using either a performance threshold $\tau$ or a memory budget. This means LUQ is only "automatic" conditional on (i) choosing a validation benchmark/metric and (ii) choosing a threshold/budget. In Sec. 4.2, LLaVA uses an MME-Perception constraint (100-point decrease relative to GPTQ-4), while Qwen uses a 4.0GB budget. The paper does not analyze how sensitive the selected layer set (and resulting accuracy on other benchmarks) is to the choice of validation metric and $\tau$. This could be a problem because it affects reproducibility and how a user would apply the method in practice.

6. The introduction frames the goal as reducing "memory and computational resources", but the evaluation reports benchmark scores and "average bitwidth" rather than actual deployment metrics (checkpoint size on disk, peak GPU memory during inference, throughput). In addition, the paper states it "quantize[s] only the language model backbone", leaving the vision encoder/projection untouched, and does not discuss KV cache memory. If the paper wants a deployment framing, it should provide at least minimal measurements or adjust the framing to focus on weight compression of the LLM backbone only.

---

> ### Author Response · Authors · 2026-03-17
> **Response to Reviewer VEdf (Part 1/2): RC1-RC3**
>
> We sincerely thank the reviewer for the thorough evaluation of our work, helping identify concrete ways to strengthen the paper. We address each requested change below.
>
> ### **RC1. Entropy as a causal driver vs. a correlate**
> This is a fair distinction. We agree that our evidence demonstrates entropy is empirically effective as a proxy rather than establishing strict causation.
>
> However, several of the suggested properties are either subsumed by entropy or unsuitable at this level of comparison. LUQ orders structurally identical transformer blocks, so architectural differences (attention vs. MLP) are held constant and cannot be used as an alternative. Activation outliers influence entropy rather than competing with it.  . Activation outliers influence entropy rather than competing with it.
>
> **The underlying intuition is straightforward:** a layer whose output collapses to a  low-entropy distribution is encoding a simpler, more compressible function, where many weight perturbations map to nearly the same output and can therefore tolerate coarser quantization. Conversely, a layer producing high-entropy outputs relies on finer distinctions among its weights, making it more sensitive to quantization noise. Carefully disentangling entropy from all possible correlates is an interesting direction for future work.
> We have softened the causal framing and added a clarifying discussion of these points in Section 1 of the revised manuscript.
>
>
> ### **RC2. "First study" claim and variance in the abstract**
>
> We appreciate this careful reading. To clarify, our intended claim was not priority over benchmarking PTQ on MLLMs, Huang et al. (2024c) indeed evaluated and documented performance collapse below 4 bits as we mention in Section 1. What we meant to convey was that ours is the first work to diagnose why this collapse occurs (via the entropy characterization of multimodal vs. text activations) and propose a method specifically designed to address it. We acknowledge the original phrasing was imprecise and have revised the abstract and Sec. 1 to "first method for ultra-low-bit MLLM quantization" to accurately reflect this.
>
> **Regarding variance:**  We agree the original abstract phrasing was imprecise. The intended meaning was that entropy values vary more across layers for multimodal tokens (variance of entropy across layers) , but the phrase "higher statistical variance and entropy" incorrectly implies a separate variance measurement. We have revised the abstract to state the finding precisely in terms of entropy alone.
> We have also tightened the language and checked for typos throughout the manuscript.
>
>
> ### **RC3. Comparison with additional PTQ methods**
> Our original comparison focused on GPTQ and AWQ as the most widely adopted PTQ baselines and the standard methods that recent works on quantization compare against.  Both have mature, well-maintained implementations that support multimodal LLMs. We note that many other PTQ methods cited in our related work, including OmniQuant, were developed and evaluated exclusively for text-only LLMs. Applying them to MLLMs is not straightforward, as it requires non-trivial adaptation of both the quantization pipeline and the multimodal evaluation infrastructure.
>
> **That said, we have now obtained results for OmniQuant and SlimLLM on a majority of the benchmarks and added them to the revised manuscript (Table 1)**. These results confirm the same trend observed with GPTQ and AWQ: reasonable performance at 4-bit, large degradation at 3-bit, and collapse at 2-bit, with LUQ significantly outperforming all methods in the ultra-low-bit regime.
> Due to GPU time constraints, we have not yet completed the evaluation of OmniQuant and SlimLLM for 2 of the 9 VQA benchmarks; results for these remaining benchmarks will be included in any final revision of the manuscript

---

> > ### Author Response · Authors · 2026-03-17
> > **Response to Reviewer VEdf (Part 2/2): RC4-RC6**
> >
> > ### **RC4. Missing standard deviations**
> > We have added standard deviations across 3 runs to Table 1, 2  of the revised manuscript. Across configurations, the standard deviations are small confirming that the performance differences reported in Table 2 are meaningful.
> >
> >
> > ### **RC5. Sensitivity to the choice of validation metric and threshold τ**
> >
> > We would like to point the reviewer to  Sec 4.3, Fig. 4 and Fig. 5 that already have the results addressing this concern. They plot the performance of LUQ on multiple benchmarks as a function of the number of quantized layers (equivalently, the average bit-width). Since any specific choice of performance threshold τ or memory budget maps directly to a point on this frontier, these figures already capture the cross-benchmark sensitivity the reviewer asks about.
> >
> > For example, the layer set chosen for LLaVA-1.5 using an MME-Perception threshold (16 layers, 2.54 avg. bits) also degrades gracefully on VQAv2 (Fig. 4b), showing that the Pareto frontiers across benchmarks are broadly consistent. To make this connection more explicit, we have added a short discussion in Sec. 4.3 on cross-benchmark transfer of the threshold choice.
> >
> > ### **RC6. Deployment metrics and scope of compression**
> >
> > We respectfully note that our paper's goal is to study the quantization robustness of multimodal LLMs in the ultra-low bit regime, not to propose a deployment system.
> >
> >
> > We report performance vs. average bit-width following the standard evaluation practice in PTQ literature (Sec 2.2: GPTQ, AWQ, BiLLM, OmniQuant)  because it decouples algorithmic compression from hardware and kernel level factors. That said, to address this concern and a similar one from Reviewer egN1, we have added throughput measurements using llama.cpp to Appendix D of the revised manuscript (please see our reply to Reviewer egN1, W2 for full details), confirming that LUQ's compression does translate to real throughput gains (~1.9× over 4-bit on an i7 CPU).
> >
> >
> > **Regarding scope:** We quantize only the LLM backbone because it constitutes the vast majority of parameters (e.g., >95% in LLaVA-1.5). KV cache quantization and vision encoder compression are orthogonal research directions with their own bodies of work, and are beyond the scope of this work.

---

### Review · Reviewer_SJKa · 2026-06-03

**Summary Of Contributions:**

I never accepted this paper to review.  I never signed up as a review of TMLR.

**Audience:**

Yes

**Audience Explanation:**

I'd like to be removed as a reviewer from TMLR.  I never signed up to be a reviewer of that journal.

**Broader Impact Concerns:**

I don't have any concerns.

**Claims And Evidence:**

Yes

**Claims Explanation:**

I did not review this paper.  I'm just trying to get the task off of my notifications by submitting this rveiew.

**Requested Changes:**

Not applicable.

---

### Decision · Action_Editor_ew3k · 2026-06-12

**Recommendation:** Accept as is

**Additional Comments:**

The reviewers are generally in agreement about the quality of the paper and although they requested a number of changes, none of them were too significant and in any case the authors have addressed the concerns well. The paper is suitable for publication.

**Audience:**

Yes

**Audience Explanation:**

The topic is salient and the investigation thorough and the community will surely find this work interesting.

**Claims And Evidence:**

Yes

**Claims Explanation:**

Yes. That activation entropy provides a good lens for quantization robustness makes intuitive sense and while the paper provides only limited support for this contention theoretically, the reviewers agree that the experimental studies provide convincing evidence and strong support for this and for the paper's main claims.